# Genome-Wide Association Study and Genomic Prediction of Fusarium Wilt Resistance in Common Bean Core Collection

**DOI:** 10.3390/ijms242015300

**Published:** 2023-10-18

**Authors:** Kenani Chiwina, Haizheng Xiong, Gehendra Bhattarai, Ryan William Dickson, Theresa Makawa Phiri, Yilin Chen, Ibtisam Alatawi, Derek Dean, Neelendra K. Joshi, Yuyan Chen, Awais Riaz, Paul Gepts, Mark Brick, Patrick F. Byrne, Howard Schwartz, James B. Ogg, Kristin Otto, Amy Fall, Jeremy Gilbert, Ainong Shi

**Affiliations:** 1Department of Horticulture, University of Arkansas, Fayetteville, AR 72701, USA; kechiwin@uark.edu (K.C.); gb005@uark.edu (G.B.); ryand@uark.edu (R.W.D.); tmakawip@uark.edu (T.M.P.); yc046@uark.edu (Y.C.); ialatawi@uark.edu (I.A.); dadean@uark.edu (D.D.); 2Department of Entomology and Plant Pathology, University of Arkansas, Fayetteville, AR 72701, USA; nkjoshi@uark.edu; 3Department of Biological Sciences, University of Arkansas, Fayetteville, AR 72701, USA; yc050@uark.edu; 4Department of Crop, Soil and Environmental Sciences, University of Arkansas, Fayetteville, AR 72701, USA; ariaz@uark.edu; 5Department of Plant Sciences, University of California, 1 Shields Avenue, Davis, CA 95616, USA; plgepts@ucdavis.edu; 6Department of Soil and Crop Sciences, Colorado State University, Fort Collins, CO 80523, USA; mark.brick@colostate.edu (M.B.); patrick.byrne@colostate.edu (P.F.B.); barry.ogg@colostate.edu (J.B.O.); amy.fall@colostate.edu (A.F.); jeremy.gilbert@colostate.edu (J.G.); 7Department of Agricultural Biology, Colorado State University, Fort Collins, CO 80523, USA; howard.schwartz@colostate.edu (H.S.); kristen.otto@colostate.edu (K.O.)

**Keywords:** common bean, genome-wide association study, fusarium wilt, genomic prediction

## Abstract

The common bean (*Phaseolus vulgaris* L.) is a globally cultivated leguminous crop. Fusarium wilt (FW), caused by *Fusarium oxysporum* f. sp. *phaseoli* (*Fop*), is a significant disease leading to substantial yield loss in common beans. Disease-resistant cultivars are recommended to counteract this. The objective of this investigation was to identify single nucleotide polymorphism (SNP) markers associated with FW resistance and to pinpoint potential resistant common bean accessions within a core collection, utilizing a panel of 157 accessions through the Genome-wide association study (GWAS) approach with TASSEL 5 and GAPIT 3. Phenotypes for *Fop* race 1 and race 4 were matched with genotypic data from 4740 SNPs of BARCBean6K_3 Infinium Bea Chips. After ranking the 157-accession panel and revealing 21 Fusarium wilt-resistant accessions, the GWAS pinpointed 16 SNPs on chromosomes Pv04, Pv05, Pv07, Pv8, and Pv09 linked to *Fop* race 1 resistance, 23 SNPs on chromosomes Pv03, Pv04, Pv05, Pv07, Pv09, Pv10, and Pv11 associated with *Fop* race 4 resistance, and 7 SNPs on chromosomes Pv04 and Pv09 correlated with both *Fop* race 1 and race 4 resistances. Furthermore, within a 30 kb flanking region of these associated SNPs, a total of 17 candidate genes were identified. Some of these genes were annotated as classical disease resistance protein/enzymes, including NB-ARC domain proteins, Leucine-rich repeat protein kinase family proteins, zinc finger family proteins, P-loopcontaining nucleoside triphosphate hydrolase superfamily, etc. Genomic prediction (GP) accuracy for *Fop* race resistances ranged from 0.26 to 0.55. This study advanced common bean genetic enhancement through marker-assisted selection (MAS) and genomic selection (GS) strategies, paving the way for improved *Fop* resistance.

## 1. Introduction

The common bean (*Phaseolus vulgaris* L.) is a diploid plant species belonging to the Fabaceae family and is ranked third among widely cultivated leguminous crops worldwide. With a haploid genome size of approximately 600 Mb, its origin dates back to Mexico around 8000 years ago, and its cultivation has expanded to Mesoamerica and the Andes over the years [1,2]. Common beans are primarily consumed as a dry food legume due to their high protein content in the grain, pulse (pods), and leaves, although tender green pods find usage in certain regions such as East Asia, Africa, and South America [3]. The consumption of common beans is on the rise globally, and they are considered a crucial component of a healthy diet, owing to their ability to mitigate the risk of various diseases, such as obesity, diabetes, and certain cancers, due to their fiber and antioxidant (phenolic) content, particularly in Africa, the Mediterranean, and the USA [4]. Furthermore, common beans boast a long storage life and have the unique ability to fix atmospheric nitrogen in the soil through nitrogen-fixing bacteria, providing economic benefits by reducing reliance on synthetic fertilizers [5]. Despite their numerous advantages, common bean crops face significant challenges caused by soil-borne diseases, including FW triggered by *Fusarium oxysporum* f. sp. *phaseoli* (*Fop*). This devastating disease substantially reduces crop productivity, resulting in substantial losses of 50–75% in common bean crops [6,7,8]. FW manifests specific symptoms such as vascular tissue discoloration, leaf chlorosis, leaf abscission, and plant death, primarily affecting lower leaves. The disease can even hamper early shoot growth, leading to stunted plant development [9,10]. The presence of chlamydospores in the soil poses significant challenges in the control and management of FW [11].

Current control methods, including chemical, cultural, and biological approaches, have limitations, such as adverse effects on soil health and increased pathogen resistance [12,13]. Developing disease-resistant cultivars and enhancing agronomic performance through breeding offer reliable and sustainable solutions, as they not only provide better disease resistance but also promote environmental friendliness, potentially reducing production costs by eliminating the need for chemicals [14]. Thus, it is beneficial to identify potential parents of resistance to support production even in the presence of *F. oxysporum* species and to allow breeding for resistance to the pathogen in the common bean [15,16]. Phenotyping and genotypic methods are frequently used to characterize plant genotypes for their resistance to abiotic and biotic stresses. Testing a germplasm collection of common bean accessions for their ability to resist *Fop* strains has been performed through an inoculation approach [17,18]. Despite its usefulness in assessing the germplasm collection of plants, this approach may not be more effective as it relies on visual scoring, making it difficult to understand molecular mechanisms triggering resistance against the pathogen in the plant [19]. Therefore, an important advanced approach to determining genetic resistance of the common bean to *Fop* through examining the genome-wide genetic variants across diverse genetic materials to understand the distribution and effects of resistance loci along with their relationships with molecular markers is required. Single nucleotide polymorphism (SNP) markers are the most popular molecular markers that are accountable for the variability in characteristics that exist among individuals [20,21,22]. It is useful to identify SNPs associated with FW resistance in common beans [18,23].

The Genome-wide association study (GWAS) has emerged as a potent tool in the domain of molecular plant breeding, enabling the identification of molecular markers and candidate genes associated with resistance to major diseases in the common bean. In recent studies, the GWAS has been employed to identify markers associated with Fusarium wilt resistance in this crop. Notably, in a study by Paulino et al., several SNPs associated with *Fop* strains were discovered on chromosomes Pv01, Pv03, Pv04, Pv05, Pv07, Pv10, and Pv11, alongside potential candidate genes related to nucleotide-binding sites and carboxy-terminal leucine-rich repeats, using a 205-member Mesoamerican Diversity Panel (MDP) [18]. Similarly, in a study by Zia et al., 14 SNPs and 14 candidate genes situated on chromosomes Pv02, Pv04, Pv07, Pv08, and Pv09 were identified, demonstrating associations with resistance bacterial wilt isolates within a 168-member USDA common bean core collection [24]. Furthermore, Shi et al. employed the GWAS to detect 18 SNPs associated with resistance to cyst nematodes (SCN, *Heterodera glycines*) in 276 Soybean germplasm accessions, with 6 SNPs linked to HG Type 2.5.7 resistance on Pv01, Pv02, Pv03, and Pv07, and 12 SNPs linked to HG Type 1.2.3.5.6.7 resistance on Pv01, Pv03, Pv06, Pv07, Pv09, Pv10, and Pv11 [25].

Beyond the GWAS, genomic prediction (GP), often referred to as genomic selection (GS), is a revolutionary breeding approach that uses information from an individual’s DNA, specifically SNPs, to predict their genetic potential for various traits, allowing breeders to select superior individuals for breeding with greater accuracy and efficiency [26,27,28]. Notably, GP has proven to offer significant advantages over conventional marker-assisted selection (MAS) in comprehending complex traits [29]. In the context of common bean diseases, GP has been employed to study root rot disease, bacterial wilt, and resistance to two soybean cyst nematode HG types, yielding considerable genomic prediction accuracy (PA). For instance, Diaz et al. reported a PA ranging from 0.7 to 0.8 for resistance to root rot disease, while Zia et al. found a PA of 0.30 to 0.56 for resistance to bacterial wilt [24,30]. Additionally, Wen et al. observed a PA range of 0.41 to 0.52 for resistance to two soybean cyst nematode HG types in the common bean. These advancements in the GWAS and GP hold great promise in accelerating the development of disease-resistant common bean varieties through enhanced understanding and selection of desirable traits [31].

In this research, a total of 157 common bean accessions were subjected to phenotyping with *Fop* race 1 and race 4, and concurrent genotyping was performed using 4740 SNPs obtained from BARCBean6K_3 Infinium BeadChips [20]. The primary aim of this investigation was to identify specific SNPs associated with *Fop* resistance in common beans and subsequently apply genomic prediction, representing the first utilization of this approach for this particular trait. These findings could serve as sources of resistance against *Fop*, thereby contributing to the development of improved resistant varieties in common bean cultivation.

## 2. Results

### 2.1. Phenotypic Variation

The resistance scores ranging from 1 to 9 for *Fop* race 1 and race 4 were assessed in each of the 157 common bean accessions (Appendix A). The mean resistance rating was 7.0 for *Fop* race 1 and 6.8 for race 4, with corresponding standard deviations (stdv) of 2.01 and 2.10, standard errors (SE) of 0.16 and 0.17, and coefficient variations (CV) of 28.80% and 30.83%, respectively (Appendix A). These findings indicate the presence of genetic variations concerning the resistance to the *Fop* races among the 157 common bean accessions.

The distribution of *Fop* race 1 and race 4 scores among the 157 common bean accessions exhibited a right-skewed pattern (Figure 1A,B), indicating a higher proportion of susceptible accessions within the panel. Twenty-one accessions demonstrated scores of 3.0 or less for either race 1 or race 4 (Table 1), indicating their resistance to at least one of the races. Specifically, among these 21 accessions, 16 were identified as highly resistant to race 1, with disease severity scores of 3.0 or below. Notably, PI 311853 from Guatemala exhibited the lowest score of 1, followed by PI 288016 from Nicaragua and PI 309877 from Costa Rica, with scores of 1.3 and 1.4, respectively, showcasing their exceptional resistance to *Fop* race 1. Similarly, 10 out of the 21 accessions demonstrated high resistance to race 4, with scores of 3 or below. Among these, PI 209482 and PI 308908 from Costa Rica, along with PI 310778 from Guatemala, exhibited the lowest score of 1, while PI 288016 scored 1.7, highlighting their heightened resistance to *Fop* race 4. Remarkably, the accessions PI 209482, PI 308908, PI 309877, PI 288016, and PI 310842 from Costa Rica and Nicaragua were the top five accessions with the highest resistance to both *Fop* race 1 and race 4. These resistant accessions hold significant potential as valuable parental lines in common bean breeding programs to enhance FW resistance.

The correlation between resistance scores to *Fop* race 1 and race 4 was found to be moderately high, with a correlation coefficient (r) of 0.51. This suggests that certain accessions exhibited resistance to both races, indicating the presence of shared resistance alleles controlling resistance to both races.

### 2.2. Population Structure and Phylogenetic Analysis

The population structure analysis revealed the presence of two distinct clusters (sub-populations) within the 157 common bean accessions. These clusters are represented by red (Q1) and blue (Q2) colors in Figure 2. The Q1 sub-population, comprising 100 accessions, was found to be the predominant cluster, accounting for 63.7% of the total accessions (Figure 2). Further details and a comprehensive view of the accessions within each cluster are depicted in Appendix A.

The phylogenetic analysis of the 21 *Fusarium*-resistant accessions revealed the presence of two distinct clusters, with accessions of the same origin closely aligned within each cluster, indicating a smaller genetic distance within these groups, signifying a shared genetic background. Furthermore, the genetic diversity analysis highlighted variations among the resistant accessions, with 15 accessions forming a larger cluster and 6 accessions forming a smaller cluster (Figure 3). These findings underscore the genetic diversity present among the resistant accessions and suggest their potential utility in breeding programs aimed at enhancing FW resistance in the common bean.

### 2.3. Association Analysis and SNP Marker Identification

The GWAS using 4740 SNPs in combination with four models (Blink, FarmCPU, GLM, and MLM) in GAPIT 3 and three models (SMR, GLM, and MLM) in TASSEL 5 identified a total of 16 and 23 SNPs associated with resistance to *Fop* race 1 and race 4, respectively (Appendix A). Each SNP was associated with either race 1 or race 4, with an LOD (logarithm of odds = −log(P)) ≥ 4.98 in one or more of the seven models (Appendix A). Notably, seven SNPs were associated with both race 1 and race 4 across all seven models. The highest associated SNPs are presented in Table 2. The Manhattan plots and QQ plots for the four models (Blink, FarmCPU, GLM, and MLM) are in GAPIT 3, and three models (SMR, GLM, and MLM) are in TASSEL 5. For instance, Figure 4 showcases the Manhattan and QQ plots of the BLINK model for *Fop* race 1 resistance and the GLM model for race 4 resistance by GAPIT 3. Notably, the QQ plots revealed deviations from the linear models (Figure 4, Appendix A), while several SNPs with large LOD values > 4.98 (Bonferroni correction threshold) or an LOD > 4.0 were observed in various models (Figure 4, Appendix A), indicative of SNPs associated with either *Fop* race 1 or race 4 resistance.

#### 2.3.1. The Associated SNPs for Race 1 Resistance

We identified a total of sixteen SNPs associated with *Fop* race 1 resistance, distributed across chromosomes Pv04, Pv05, Pv07, Pv08, and Pv09 (Appendix A). Among these, six highly associated SNPs, namely ss715650990_Chr04_26314820, ss715647361_Chr04_45301836, ss715647824_Chr05_275140, ss715645682_Chr07_517953, ss715646092_Chr08_57870335, and ss715646367_Chr09_29788600, were located at specific genomic positions on Pv04, Pv05, Pv07, Pv08, and Pv09, respectively (Table 2).

Notably, the SNP marker ss715646367_Chr09_29788600 exhibited the highest LOD value of 11.56 based on the BLINK model and had an LOD > 5 across five models, indicating a strong association with the *Fop* race 1 resistance, with a QTL (Quantitative Trait Locus) region likely located near this SNP on Pv09. On the other hand, the SNP marker ss715647361_Chr04_45301836 showed an LOD value of 8.24 based on the BLINK model but had lower LOD values (<2.0) in five models, suggesting it might not be a stable or reliable marker.

The SNP marker ss715645682_Chr07_517953 had an LOD > 4.98 (Bonferroni correction threshold) in two models and an LOD > 4.8 across four models, and its nearby SNP, ss715645685_Chr07_606814, showed similar LOD values across all seven models and a high LOD (10.08) in *t*-test (Appendix A), indicating a potential QTL region for race 1 resistance located near these two SNPs on Pv07. Moreover, four SNPs, ss715650990_Chr04_26314820, ss715650115_Chr04_27464228, ss715650468_Chr04_27690714, and ss715649688_Chr04_27781623, located within a 1.5 Mbp region from 26,314,820 bp to 27,690,714 bp on Pv04, had an LOD > 4.8 across four models, but their R2 values were relatively low, around 2.5% in the MLM model, suggesting a QTL in this region with a modest effect on race 1 resistance. Additionally, several other SNPs exhibited an LOD > 4.89 in one or more models or an LOD > 4.0 in two or more models (Table 2 and Appendix A), indicating their association with *Fop* race 1 resistance.

Overall, our GWAS analysis revealed the presence of several significant SNPs associated with *Fop* race 1 resistance in the common bean population, providing valuable insights into the genetic basis of resistance to FW in this crop.

#### 2.3.2. The Associated SNPs for Race 4 Resistance

A total of twenty-three SNPs were found to be associated with *Fop* race 4 resistance, distributed across chromosomes Pv03, Pv04, Pv05, Pv07, Pv09, Pv10, and Pv11 (Appendix A). Among these, six SNPs, namely ss715649363_Chr03_35509497, ss715650990_Chr04_26314820, ss715645397_Chr05_37965834, ss715646025_Chr07_48806850, ss715645623_Chr09_32650091, and ss715647542_Chr11_44755455, were identified as highly associated with race 4 resistance, positioned at specific genomic locations on Pv03, Pv04, Pv05, Pv07, Pv09, and Pv11, respectively (Table 2).

The BLINK model indicated that only one SNP, ss715645623_Chr09_32650091, had an LOD > 4.98 (Bonferroni correction threshold), suggesting its strong association with *Fop* race 4 resistance. Similarly, based on the FarmCPU model, only ss715647542_Chr11_44755455 showed an LOD > 4.98 (Table 2 and Appendix A), indicating its significant association with *Fop* race 4 resistance. 

For Pv03, the SNP marker ss715649363_Chr03_35509497 had an LOD > 4.98 across three models, and two adjacent SNPs, ss715650616_Chr03_32369163 and ss715647848_Chr03_33503791, had LOD values > 4.98 and >4.0 in two models, respectively, suggesting the presence of a QTL region for race 4 resistance spanning a length of 3.15 Mbp from 32,369,163 bp to 35,509,497 bp on Pv03.

On Pv04, four SNPs, ss715651182_Chr04_11640123, ss715648302_Chr04_12157925, ss715648999_Chr04_12447195, and ss715649810_Chr04_13397055, extended over a 1.76 Mbp region from 11,640,123 bp to 13,397,055 bp and exhibited an LOD > 4.5 across three models, indicating another QTL region for race 4 resistance. Additionally, a second QTL region on Pv04 was identified, represented by the SNPs ss715650115_Chr04_27464228, ss715650468_Chr04_27690714, and ss715649688_Chr04_27781623, spanning 320 Kb from 27,464,228 bp to 27,781,623 bp and having an LOD > 5.7 in three models.

On Pv07, the SNP marker ss715646025_Chr07_48806850, along with two nearby SNPs, ss715648570_Chr07_48450279 and ss715646020_Chr07_48927436, extended over 478 Kb from 48,450,279 bp to 48,927,436 bp, and exhibited an LOD ≥ 4.98 in two or three models, suggesting the presence of a QTL region for race 4 resistance in this region (Appendix A).

Moreover, other SNPs listed in Table 2 and Appendix A showed an LOD > 4.98 in one or more models, indicating potential QTL regions for race 4 resistance in their vicinity. The presence of multiple SNPs associated with *Fop* race 4 resistance highlights the genetic complexity underlying the resistance traits in the common bean population.

Seven SNPs, encompassing five on Pv04 and two on Pv09, were found to be consistently associated with resistance to both *Fop* races’ pathogens (Appendix A). This simultaneous association suggests the presence of shared genomic regions influencing resistance to both races. Among the five SNPs on Pv04, namely ss715650990_Chr04_26314820, ss715650115_Chr04_27464228, ss715650468_Chr04_27690714, and ss715649688_Chr04_27781623, a genomic segment spanning 1.5 Mbp (from 26,314,820 bp to 27,781,623 bp) exhibited LOD scores greater than 4.6 across four models, indicating the presence of a QTL in this region on Pv04. Similarly, the two SNPs, ss715648883_Chr09_22785976 and ss715646055_Chr09_25385192, located within a 2.6 Mbp interval (from 22,785,976 bp to 25,385,192 bp) on Pv09, showed LOD scores of at least 4.0 across four models, suggesting the presence of a QTL in this region on Pv09 (Appendix A).

### 2.4. Candidate Genes for Fusarium Wilt Resistance

Within a 30 kb distance from the 16 associated SNPs for *Fop* race 1 resistance in Appendix A and 23 SNPs for race 4 resistance in Appendix A, a total of 153 genes were identified (Appendix A). Among these genes, ten were associated with race 1 and 7 with race 4 resistances (Table 3).

Notably, *Phvul.004G016532*, located on chromosome Pv04, contained an NB-ARC domain-containing disease resistance protein and was positioned within 30 kb of the SNP marker ss715647806_Chr04_1827663, which was associated with *Fop* race 1 resistance. Additionally, *Phvul.009G195900* on Pv09 and *Phvul.009G195901* on Pv03 were found near the SNPs ss715646367_Chr09_29788600 and ss715650616_Chr03_32369163, respectively. These genes contained Leucine-rich repeat protein kinase family proteins and were associated with *Fop* race 1 and race 4 resistance, respectively. Another gene, *Phvul.004G151100*, located near SNP ss715647361_Chr04_45301836 on Pv04, was found to contain a Zinc finger and P-loop superfamily protein, suggesting its potential role in *Fop* resistance (Table 3).

While the 17 genes identified in this study present promising candidates for FW resistance, further assessments and validation are necessary to confirm their actual association with the resistance traits. These candidate genes hold significant potential for enhancing our understanding of the molecular mechanisms underlying FW resistance in the common bean.

### 2.5. Genomic Prediction

The GP used five models in combination with three sets of SNPs. The PA varied from 0.26 to 0.29 for the set of all 4740 SNPs and 0.42 to 0.47 for the GWAS-derived 32 associated SNPs as opposed to the low PA from 0.01 to 0.14 for the randomly selected 32 SNPs for the *Fop* race 1 resistance across the five models. Correspondingly, the PA ranged from 0.31 to 0.34 for all 4740 SNPs and 0.53 to 0.55 for the 32 associated SNPs as compared to a low PA of 0.05 to 0.24 for the 32 randomly selected SNPs for the *Fop* race 4 resistance (Figure 5 right, Appendix A). The five GP models had similar PA in each of the three SNP sets, indicating each of the five GP models can be used in GS to select *Fop* resistance.

Among the three SNP sets, the set “m32” of the GWAS-derived 32 associated SNPs had the highest PA with a mean of 0.45 for *Fop* race 1 and 0.54 for race 4 resistance estimated from five models; the “4740SNP” of all 4740 SNPs had the second highest, with a mean of 0.28 for *Fop* race 1 and 0.32 for race 4 resistance; and the “r32” set of randomly selected 32 SNPs had the lowest, with a mean of 0.10 for *Fop* race 1 and 0.16 for race 4 resistance (Figure 5, Appendix A), showing that the GWAS derived markers can be utilized in GS for selecting FW resistance in the common bean. 

## 3. Discussion

### 3.1. Genetic Diversity and Population Structure for the Common Bean Germplasm

The genetic diversity of the common bean has been extensively investigated across diverse regions worldwide, employing various approaches, such as morphological characterization [32], simple sequence repeats (SSR) [33], high-density SNPs [34,35], and allozymes [36] to comprehensively assess the genetic variability and population structure of the common bean germplasm [20]. Notably, the common bean, as a significant legume crop, exhibits noteworthy variation in crucial traits, including days to flowering, days to maturity, number of pods per plant, and seed-related characteristics [30]. The presence of two distinct gene pools, namely the Mesoamerican and Andean gene pools, has been unequivocally confirmed, providing insights into the evolutionary and domestication processes of this crop. Furthermore, the representativity of core collections of the common bean germplasm has undergone thorough evaluation through SNP diversity assessment, underlining the significance of genetic diversity evaluation for advancing crop improvement strategies. In this context, the genetic diversity and population structure results from our study align with previous findings, reinforcing the notion of an optimal genetic diversity status for the common bean [20,37].

### 3.2. Fusarium Wilt Phenotyping

Fusarium wilt is a destructive disease affecting various crops such as the common bean [14,15], banana [38], tomato [39], cowpea [40], and chickpea [41]. Phenotyping is essential for accurately assessing disease severity and identifying resistant genotypes. The significance of phenotyping lies in its role in breeding programs and disease management strategies [13]. By evaluating disease severity in different genotypes, researchers can select resistant plants for further breeding efforts [42,43]. Phenotyping also aids in understanding the genetic basis of resistance and studying host–pathogen interactions [8]. However, FW phenotyping faces challenges due to environmental influences and the lack of standardized protocols [44]. Variability in disease severity across locations and seasons can affect data comparability, while subjective visual rating scales may introduce bias. To address these challenges, advanced techniques have been explored. Quantitative phenotyping metrics, such as stem vascular discoloration length and the number of *Fusarium* necrotic vessels, offer objective and reproducible assessments of vascular damage [45]. In this study, the detection method employed for FW phenotyping was comprehensive, integrating symptoms observed in both leaves and stems, including drying, wilting, and chlorosis. By considering a wider range of symptoms, the researchers aimed to enhance the accuracy and stability of ranking for the resistance level based on disease severity. It enhances the understanding of the disease’s impact on the plant and contributes to the development of more effective strategies for combating this destructive pathogen in various crops [46].

### 3.3. Genome-Wide Association Study and SNP Identification

The present study identified SNPs associated with FW resistance in the common bean, with these markers distributed across various chromosomes, including Pv03, Pv04, Pv05, Pv07, Pv08, Pv09, Pv10, and Pv11. Notably, seven SNPs on chromosomes Pv04 and Pv09 were found to be concurrently linked to resistance against both *Fop* race 1 and race 4. These regions associated with *Fusarium* resistance align with those documented by Paulino et al., who conducted a GWAS on *Fusarium* resistance in a core collection comprising 205 common bean genotypes sourced from the germplasm bank at the Agronomic Institute (IAC, Campinas, SP, Brazil) [18].

Moreover, Leitão et al. performed association mapping using 162 Portuguese genotypes of the common bean and identified nine significant SNPs associated with FW resistance on chromosomes Pv04, Pv05, Pv07, and Pv08 [23]. Intriguingly, these same chromosomes were reported to be associated with resistance to other diseases in the common bean in studies by Perseguini et al. and Zia et al. [24,47]. Perseguini found significant associations for resistance to anthracnose and angular leaf spot on chromosomes Pv03, Pv04, and Pv08, while Zia discovered fourteen SNP associations for resistance to bacterial wilt on chromosomes Pv02, Pv04, Pv07, Pv08, Pv10, and Pv11. Similarly, a GWAS analysis by Monteiro et al. indicated ten SNPs on chromosomes Pv01, Pv03, Pv06, Pv07, Pv08, Pv09, Pv10, and Pv01 linked to resistance against *Xanthomonas citri* pv. *fuscans* in *Phaseolus vulgaris* [48]. These findings collectively demonstrate that these SNP associations confer resistance against a diverse range of pathogens.

The significant SNPs identified in this study that regulate FW resistance hold considerable promise for enhancing elite cultivars through marker-assisted selection breeding programs. By leveraging these markers, breeders can efficiently and accurately select for *Fusarium*-resistant genotypes, facilitating the development of improved common bean varieties with enhanced disease resistance. This approach not only accelerates the breeding process but also ensures the preservation and utilization of valuable genetic resources in crop improvement efforts. The comprehensive understanding of SNP associations with disease resistance contributes to the development of sustainable disease management strategies, further reinforcing the significance of this study’s findings in common bean breeding programs [49].

Additionally, the differences in the GWAS results arising from various GWAS models stem from variations in statistical methods, data preprocessing, genetic models, marker density, sample size and diversity, phenotype definition, software and algorithms, and interpretation. These variations can influence the identification of significant genetic associations, the number of detected markers, and the power to detect specific loci. In this study, we employed meta-analyses that amalgamated findings from a variety of GWAS models, offering a more comprehensive and resilient perspective on genetic associations [50].

### 3.4. Candidate Gene for Fusarium Wilt Resistance

The characterization of genes involved in disease resistance mechanisms is crucial for a comprehensive understanding of plant defense responses. This study identified seventeen candidate genes, including *Phvul.004G016532*, *Phvul.004G151100*, and others, which were found to contain receptors such as NB-ARC domain-containing disease resistance protein [51], zinc finger (Ran-binding) family protein [52], and leucine-rich repeat protein kinase family protein [53], and P-loop-containing nucleoside triphosphate hydrolase superfamily proteins. These findings align with previous reports on leguminous plants such as soybean [54], cowpea [55], and chickpea [56]. The NB-ARC domain is responsible for the ATP or GTP binding and hydrolysis activity that is crucial for signal transduction in plant immune responses [51,57]. Leucine-rich repeat protein kinase family proteins, as indicated by Schmidt et al., are essential mediators of cell-to-cell interaction, transmitting developmental signals and environmental stimuli and triggering defense or resistance against pathogens [58]. Similarly, the zinc finger family proteins identified in this study are known to participate in diverse metabolic pathways and contribute to stress response and defense against pathogens in plants, particularly associated with the Jasmonic acid-dependent pathway [59]. And, the P-loop-containing nucleoside triphosphate hydrolase superfamily proteins in plants are vital components of disease resistance mechanisms, facilitating nucleotide binding and hydrolysis to initiate immune responses upon pathogen detection [60]. The model genes and their associated SNPs identified in this study offer valuable markers for successful marker-assisted breeding in the common bean, facilitating the development of disease-resistant cultivars and contributing to the improvement in crop productivity.

### 3.5. Genomic Prediction for Genomic Selection of Fusarium Wilt Resistance

The genomic prediction of disease resistance in legumes has emerged as a powerful tool in modern agricultural research and crop improvement. The constant threat posed by various pathogens to legume crops has driven the need for more effective and sustainable approaches to combat diseases. Genomic prediction leverages advancements in genomics and computational biology to enhance disease resistance in legumes through targeted breeding strategies. The references provided offer insights into this promising field, shedding light on the potential benefits and challenges associated with utilizing genomic information for disease resistance prediction in legumes [61]. The research conducted by Keller et al. centered around the common bean and unveiled significant findings concerning the multifaceted trials emanating from environmental stressors, encompassing maladies, drought-induced pressures, and restricted phosphorus availability. In this context, the domain of genomic prediction arises as a promising solution, leveraging genomic data to forecast crucial agronomic traits, particularly including diseases [62].

The current study, the first report of genomic prediction in FW resistance in the common bean, was conducted using three sets of SNPs (m32, r32, and 4740SNPs) with the application of five genomic prediction models (rrBLUP, BA, BB, BRR, and BL). The highest PA values were achieved when utilizing the 32 GWAS-derived associated SNPs, followed by all 4740 SNPs, for both *Fop* race resistances. Analysis of the five prediction models revealed a consistent trend in PA for both *Fop* race 1 and race 4 resistances (Appendix A; Figure 4), indicating that the GWAS-derived SNP set led to enhanced PA values, and increasing the number of SNPs further augmented the PA value. This trend has been observed in prior studies involving different plant traits. Ravelombola et al. reported PA exceeding 0.5 (50%) in the set of GWAS-derived SNPs associated with reduced soybean chlorophyll content and soybean cyst nematode tolerance [63]. Similarly, Shi et al. demonstrated GWAS-derived SNP sets with PA greater than 0.7 for white rust resistance in the USDA GRIN spinach germplasm [64]. These findings underscore the potential utility and effectiveness of employing GWAS-derived SNP sets in enhancing the predictive ability of genomic prediction models for disease resistance traits in various plant species.

The consistent trends observed in different GP models across various datasets of disease resistance can be attributed to the underlying genetic architecture of the trait, which often involves a combination of common genetic markers affecting resistance. These shared markers, while specific to different datasets, may be indicative of core genetic factors that play a significant role in disease resistance across diverse populations, resulting in similar model predictions despite dataset variations [65,66]. Additionally, the robustness and adaptability of GP models can contribute to their ability to capture these common genetic patterns and produce consistent trends in different datasets, further emphasizing the relevance and reliability of GP in disease resistance prediction for common beans.

## 4. Materials and Method

### 4.1. Plant Material and Phenotyping

The 157 accessions of the common bean, obtained from the USDA/ARS Western Regional Plant Introduction Station, Pullman, WA, germplasm collection, which originated from ten countries, were evaluated for resistance to *F. oxysporum* races 1 and 4 in Fort Collins, CO, in a controlled greenhouse by Dr. Brick and collaborators [17]. Under controlled greenhouse conditions, the study maintained a temperature of approximately 16 °C during the night and 32 °C during the day, supplemented with artificial lighting for 13 h daily. The root dip inoculation procedure was employed to screen seedlings from each of the 157 common bean accessions. After 21 days of inoculation, the plants were assessed for their reaction to *Fusarium oxysporum* f. sp. *phaseoli* (*Fop*) using the CIAT disease severity scale [67]. This scale, also known as the severity index, categorized the plants based on the percentage of leaf tissue exhibiting disease symptoms, including drying, wilting, or chlorosis, as follows: 1 = no disease symptoms and completely healthy; 3 = 10% of the leaf surface area showing disease symptoms; 5 = 25% of the leaf surface showing disease symptoms along with some whole plant stunting; 7 = disease symptoms on 50% of leaves and severely stunted; and 9 = plant death. According to the CIAT disease severity score, plants were classified as resistant (score 1–3), intermediate (score 4–6), or susceptible (score 7–9). A total of 8 to 10 seedlings from each accession were evaluated, and the average severity index (ASI) was calculated for each accession based on all evaluated plants. To validate the pathogenicity and confirm disease classification, both resistant and susceptible check entries were included in the experiments. Cultivar UI 114 served as the susceptible check (ASI > 8), while the line Lef-2RB consistently acted as the resistant check (ASI < 3). Additionally, two non-inoculated plants from each accession were grown to determine the phenotype in the absence of disease symptoms and evaluate the pathogenicity of the inoculum alongside the inoculated plants. The phenotyping data analysis for the two *Fop* races was conducted using ANOVA functions in JMP Genomics 7 (SAS Institute Inc., Cary, NC, 1989–2023). The parameters were estimated for the mean, variance, standard deviation, and standard error. These parameters were evaluated using the “Tabulate” function in JMP Genomics 7, and the distribution function was used to graphically present the phenotyping data for each of the *Fop* races.

### 4.2. Genotyping and SNP Selection

A set of 157 common bean accessions was genotyped using BARCBean6K_3 Infinium Bead Chips, and SNPs across the 157 accessions were downloaded from the SNP dataset at (https://datadryad.org/stash/dataset/doi:10.25338/B8KP45, accessed on 23 February 2023) [20]. For enhancing statistical power and quality control, the SNPs were filtered with the exclusion of SNPs: data missing rate > 20%, heterogeneous > 10%, and MAF (minor allele frequency) < 5%. After filtering, 4740 SNPs distributed on the 11 chromosomes (Appendix A) were used for GWAS of *Fop* resistance in this study.

### 4.3. Principal Component Analysis (PCA) and Genetic Diversity

Principal component analysis (PCA) and genetic diversity analysis were conducted using 4740 SNPs in GAPIT 3 (https://github.com/jiabowang/GAPIT3, accessed on 23 February 2023), with PCA set to range from 2 to 10. Additionally, phylogenetic trees were generated using the neighbor-joining (NJ) method [68] and were drawn by MEGA 11 [69].

### 4.4. Genome-Wide Association Study and SNP Marker Identification

Genome-wide association mapping was performed for the 157 accessions of common bean using various statistical models, including Bayesian-information and Linkage-disequilibrium Iteratively Nested Keyway (BLINK) [70], mixed linear model (MLM) [71], general linear model (GLM) [72], Fixed and random model Circulating Probability Unification (FarmCPU) [73] in GAPIT 3 (https://zzlab.net/GAPIT/index.html, accessed on 23 February 2023), as well as single marker regression (SMR) [74], generalized linear model (GLM), and mixed linear model (MLM) methods in TASSEL 5 (https://www.maizegenetics.net/tassel, accessed on 23 February 2023) [75]. Manhattan plots and QQ plots for all association models were generated using GAPIT 3 and TASSEL 5. The linkage disequilibrium (LD) between markers was calculated using the squared correlation coefficient (R^2^) to help ensure the accuracy and reliability of GWAS.

### 4.5. Candidate Gene Estimation

All SNP loci significantly associated with either *Fop* race 1 or race 4 were subjected to candidate gene prediction for the discovery of candidate genes covering the 50 kb regions. The Andean whole-genome reference sequence *Pvulgaris* 442_v2.1 presented on the Phytozome website (https://phytozome.jgi.doe.gov/pz/portal.html, accessed on 28 February 2023) was explored to retrieve the candidate genes from the reference annotation of the common bean genome.

### 4.6. Genomic Prediction for Genomic Selection of Fusarium Wilt Resistance

GP was performed to analyze the effect of the SNPs identified in the association analysis using five different models: best linear unbiased prediction (BLUP) [76] and Bayesian models Bayes A (BA), Bayes B (BB), Bayes ridge regression (BRR), and Bayes LASSO (BL) [77] (Appendix A). For validating the performance of the identified markers of FW resistance, three sets of SNPs, m32 (32 associated SNPs), r32 (randomly selected 32 SNPs), and 4740 (all 4740 SNPs), were used along with the aforementioned five models. The distribution plots were generated using RStudio (R version 4.2.2). For each GP model, a five-fold cross-validation was conducted [77]. This approach involves dividing the dataset into multiple subsets, iteratively training the predictive model on a portion of the data and evaluating its performance on the remaining data to assess its accuracy and generalizability. The association panel was randomly partitioned into five non-overlapping subsets, with four subsets utilized as the training set and the remaining subset as the testing set. This process was replicated 100 times for each fold. Mean and standard errors were then computed for each fold. Genomic prediction accuracy (PA) was assessed by calculating the Pearson’s correlation coefficient (r) between the Genomic Estimated Breeding Values (GEBV, a genetic parameter to predict the potential of individuals for specific traits) and the observed phenotypic values for the testing set, following the methodology described by Shikha et al. [78].

## 5. Conclusions

This study utilized the GWAS and GP to investigate Fusarium wilt resistance in 157 USDA common bean accessions, identifying resistant accessions and SNP markers associated with resistance with potential applications in targeted breeding. Among the accessions assessed, twenty-one were found to exhibit high resistance to either *Fop* race 1 or race 4, with a disease score ≤ 3 (not more than 10% of the leaf surface area showing disease symptoms). Additionally, five USDA common bean accessions, PI 209482, PI 308908, PI 309877, PI 288016, and PI 310842, were identified as resistant to both races, signifying their potential to develop resistant lines. Furthermore, the study identified thirty-nine SNPs located on chromosomes Pv03, Pv04, Pv05, Pv07, Pv08, Pv09, Pv10, and Pv11, along with seventeen candidate genes associated with *Fop* resistance. GP was conducted for *Fop* race 1 and race 4 resistance, and PA ranging from 0.26 to 0.55 was observed, indicating the potential for predictive accuracy. The identified SNP markers and associated genes are powerful tools in plant breeding, enabling targeted selection of Fusarium-resistant plants, development of genotyping assays for large populations, and the introgression of resistance alleles from diverse germplasm, ultimately enhancing breeding programs’ efficiency and precision. In summary, the findings from this study of common bean Fusarium wilt resistance hold significant promise for farmers and crop improvement programs. They enable the development of resistant cultivars, reduce reliance on chemical pesticides, enhance crop yield and quality, and contribute to sustainable and economically viable bean production.

## Figures and Tables

**Figure 1 ijms-24-15300-f001:**
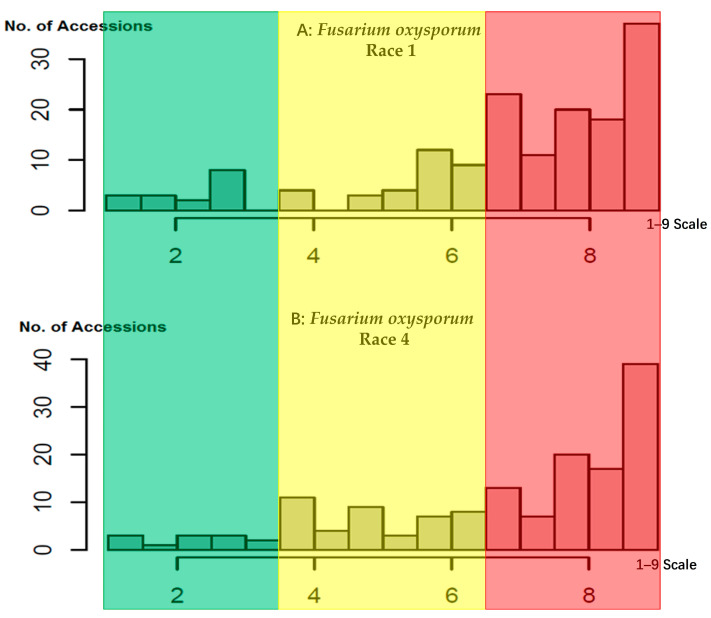
Distribution of *Fusarium oxysporum* (*Fop*) rating (1–9 scale) in 157 common bean accessions. Green: resistance, yellow: medium resistance, and red: susceptibility. (**A**) *Fop* race 1 and (**B**) *Fop* race 4.

**Figure 2 ijms-24-15300-f002:**
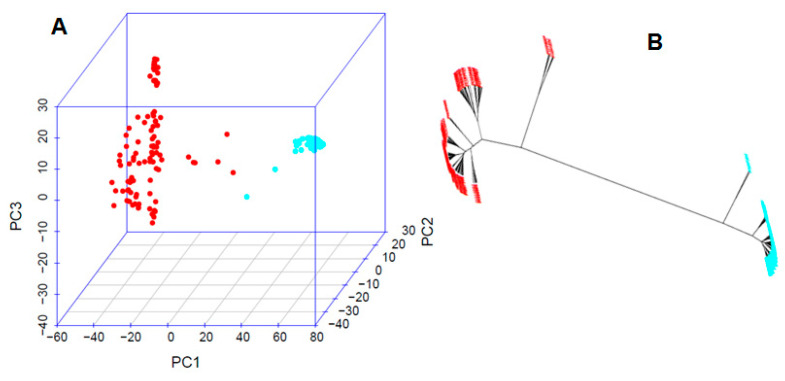
Population structure for the 157-common bean panel. (**A**) The 3D graphical plot of the principal component analysis (PCA) and (**B**) phylogenetic trees created using the neighbor-joining (NJ) method in two subpopulations created using GAPIT 3. The distribution of the accessions to different populations is indicated by the color code (Q1: red and Q2: blue).

**Figure 3 ijms-24-15300-f003:**
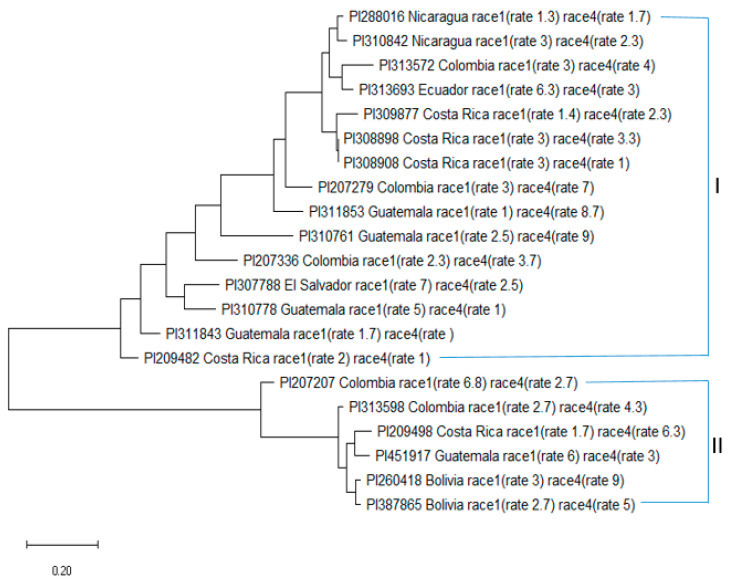
Phylogenetic tree in 21 Fusarium wilt-resistant common bean accessions, the I and II represented the two sub-poulations (Q1 and Q2) from the structure analysis, respectively.

**Figure 4 ijms-24-15300-f004:**
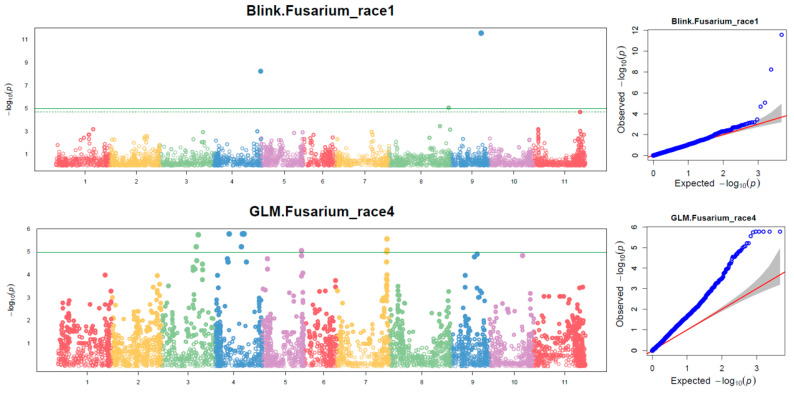
The Manhattan (**Left**) and QQ (**Right**) plots of BLINK model for Fusarium wilt race 1 resistance and GLM model for FW race 4 resistances by GAPIT 3. The vertical and horizontal axes represent observed vs. expected logarithm of odds (LOD or −log(*p*-value)).

**Figure 5 ijms-24-15300-f005:**
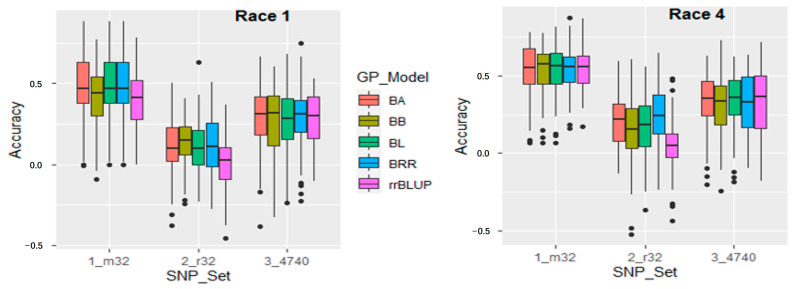
Genomic prediction (r-value) in the y-axis for resistance to Fusarium wilt race 1 and race 4 in 157 common bean accessions estimated by five GP models: BA, BB, BL, BRR, and rrBLUP on three SNP sets (x-axis), 1_m32 (32 associated SNPs), 2_r32 (randomly selected 32 SNPs), and 3_4740 (all 4740 SNPs).

**Table 1 ijms-24-15300-t001:** The 21 Fusarium wilt (race1/race4) resistance common bean accessions.

PI_Accession	PI_Name	Origin.Country	Race1_2006_ Disease_Rate	Race4_2006_ Disease_Rate	Rate ≤ 3 Either Race 1 or Race 4 or Both
PI260418	PV-3	Bolivia	3	9	race 1 ≤ 3
PI387865	W-941d	Bolivia	2.7	5	race 1 ≤ 3
PI207207	Z-#4	Colombia	6.8	2.7	race 4 ≤ 3
PI207279	Chiapas 36-3	Colombia	3	7	race 1 ≤ 3
PI207336	Jalisco 31-1	Colombia	2.3	3.7	race 1 ≤ 3
PI313572	Antioquia 12	Colombia	3	4	race 1 ≤ 3
PI313598	Cauca 38	Colombia	2.7	4.3	race 1 ≤ 3
PI209482	G16837	Costa Rica	2	1	race 1 and 4 ≤ 3
PI209498	G1363	Costa Rica	1.7	6.3	race 1 ≤ 3
PI308898	Line 7	Costa Rica	3	3.3	race 1 ≤ 3
PI308908	Criollo blanco No. 2	Costa Rica	3	1	race 1 and 4 ≤ 3
PI309877	Col. No. 20670, lot #33	Costa Rica	1.4	2.3	race 1 and 4 ≤ 3
PI313693	Col. No.	Ecuador	6.3	3	race 4 ≤ 3
PI307788	S-219-R	El Salvador	7	2.5	race 4 ≤ 3
PI310761	G2022	Guatemala	2.5	9	race 1 ≤ 3
PI310778	G2031	Guatemala	5	1	race 4 ≤ 3
PI311843	Frijol de gato	Guatemala	1.7		race 1 ≤ 3
PI311853	Colorado del suelo	Guatemala	1	8.7	race 1 ≤ 3
PI451917		Guatemala	6	3	race 4 ≤ 3
PI288016	Negro Nicaraguense	Nicaragua	1.3	1.7	race 1 and 4 ≤ 3
PI310842	G2084	Nicaragua	3	2.3	race 1 and 4 ≤ 3

**Table 2 ijms-24-15300-t002:** SNPs associated with resistance to Fusarium wilt race 1 and race 4 based on seven models.

SNP	Chr	Position (bp)	LOD [−log(*p*-Value)]								Associated FW Race
			GAPIT 3				TASSEL 5			*t*-Test	
			Blink	FarmCPU	GLM	MLM	SMR	GLM	MLM		
ss715650990_Chr04_26314820	4	26,314,820	0.47	2.92	4.87	4.63	4.87	4.63	1.30	3.92	Race 1
ss715647361_Chr04_45301836	4	45,301,836	8.24	0.79	1.66	1.26	1.66	1.26	2.54	1.97	
ss715647824_Chr05_275140	5	275,140	2.40	3.22	5.04	4.77	5.04	4.77	2.13	3.76	
ss715645682_Chr07_517953	7	517,953	1.07	3.04	5.07	4.83	5.07	4.83	2.09	10.08	
ss715646092_Chr08_57870335	8	57,870,335	5.06	2.15	3.13	2.74	3.13	2.74	1.40	3.21	
ss715646367_Chr09_29788600	9	29,788,600	11.56	3.30	5.52	5.23	5.52	5.23	2.46	3.42	
ss715649363_Chr03_35509497	3	35,509,497	2.78	2.85	5.73	2.34	5.59	6.11	2.10	4.16	Race 4
ss715650990_Chr04_26314820	4	26,314,820	3.55	1.60	5.77	2.81	6.42	6.69	2.81	7.26	
ss715645397_Chr05_37965834	5	37,965,834	1.64	4.77	5.04	2.62	3.58	4.90	2.02	4.45	
ss715646025_Chr07_48806850	7	48,806,850	2.05	0.89	5.55	2.14	5.76	5.87	1.94	4.59	
ss715645623_Chr09_32650091	9	32,650,091	5.05	4.18	2.85	2.75	1.93	2.70	2.25	5.50	
ss715647542_Chr11_44755455	11	44,755,455	2.72	6.01	2.39	2.81	0.01	2.50	2.84	0.31	

**Table 3 ijms-24-15300-t003:** Seventeen candidate genes, including five disease resistance genes from the associated SNPs, where 10 genes are for Fusarium wilt race 1 resistance and 7 genes are for race 4 resistance.

Gene	Chr	Gene Size (bp)	Gene Annotation	Comment	Closest SNP Marker	Fusarium Wilt Race	Distance (kb)
*Phvul.004G016532*	4	4457	NB-ARC domain-containing disease resistance protein	R gene	ss715647806_Chr04_1827663	race1	<30 kb
*Phvul.004G016800*	4	5948	HD domain-containing metal-dependent phosphohydrolase family protein		ss715647808_Chr04_1845589		<1 kb
*Phvul.004G151100*	4	6906	Zinc finger (Ran-binding) family protein	R gene	ss715647361_Chr04_45301836		0
*Phvul.005G003400*	5	2088	SNARE-like superfamily protein		ss715647824_Chr05_275140		<2 kb
*Phvul.007G007100*	7	3311	Nitrilase/cyanide hydratase and apolipoprotein N-acyltransferase family protein		ss715645682_Chr07_517953		<2 kb
*Phvul.007G008400*	7	2702	Peroxidase superfamily protein		ss715645685_Chr07_606814		<1 kb
*Phvul.007G008300*	7	3835	Pre-mRNA-splicing factor 3				<2 kb
*Phvul.008G228500*	8	4138	Protein of Unknown Function (DUF239)		ss715646092_Chr08_57870335		0
*Phvul.009G153600*	9	7435	Amino acid permease 2		ss715648883_Chr09_22785976		0
*Phvul.009G195900*	9	2567	Leucine-rich repeat protein kinase family protein	R gene	ss715646367_Chr09_29788600		<10 kb
*Phvul.003G129400*	3	2678	Leucine-rich repeat protein kinase family protein	R gene	ss715650616_Chr03_32369163	race4	<5 kb
*Phvul.004G073900*	4	13,305	Polynucleotidyl transferase, ribonuclease H fold protein with HRDC domain		ss715648302_Chr04_12157925		<2 kb
*Phvul.005G045400*	5	2233	P-loop-containing nucleoside triphosphate hydrolase superfamily protein	R gene	ss715650411_Chr05_4808704		<3 kb
*Phvul.005G137400*	5	4765	Carbamoyl phosphate synthetase B		ss715645397_Chr05_37965834		<2 kb
*Phvul.009G153600*	9	7435	Amino acid permease 2		ss715648883_Chr09_22785976		0
*Phvul.009G216500*	9	18,256	RNA binding		ss715645623_Chr09_32650091		<1 kb
*Phvul.010G071766*	10	6447	Nucleotidyltransferase family protein		ss715650855_Chr10_32133091		0

## Data Availability

The original information presented in the study is available in the article/Appendix A.

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
