# Peer review of "Genome-Wide Association Study and Genomic Prediction of Fusarium Wilt Resistance in Common Bean Core Collection"

_ijms, 2023, doi:10.3390/ijms242015300_

Round 1
Reviewer 1 Report
Fusarium oxysporum f. sp. phaseoli causes a very serious wilt disease of common beans worldwide. Due to the harmfulness of fungicides to humans and the environment, searching for and breeding resistant cultivars is an important alternative. In this manuscript, the 157 accessions of the common bean, which originated from ten countries, were evaluated for resistance to two races of F. oxysporum. All chapters are written at a high substantive level. The individual stages of research are presented in detail and clearly in M&M. The results are precisely presented in tables and figures. There is relatively much documentation in the Supplementary. The discussion is very interesting. After minor correction, the manuscript should be published in IJMS/MDPI. Because there are some mistakes and typographiacal errors (see Remarks), it is advisable to read the entire manuscript carefully and check the correctness of the text.
Remarks
Line 41 regions[3] – add a space. Such errors are numerous and should be corrected throughout the manuscript.
Figure 1 - It should be marked in Figure 1 where A is and where B is. Apart from that, what does FW (why not Fop) mean; race concerns the fungus Fusarium oxysporum ... and not the disease Fusarium wilt
Line 153 Fusarium – it should be in italic, also in some other places in the manuscript where the pathogen is mentioned (of course, this does not apply to Fusarium wilt disease)
Line 186 it is ‘SNP markers for race 4 resistance’ – it should be ‘SNP markers for race 1 resistance’
Line 218 it is ‘with Fop race 2’ - it should be rather with Fop race 4
Table 2, it is unclear which positions apply to race 1 and which to race 2 (see text in Line 175)
Table 3 similarly as in Table 2, it is unclear which positions apply to race 1 and which to race 2 (see text in Lines 272-273)
Line 290 table 3 – misconstructed table description – the verb is missing
Line 357 Paulino et al.- in ‘Literature’ you write de Carvalho Paulino; this should be consistent
Line 360 Leita˜o - in ‘Literature’ you write Leitao,
Line 364 you write Persenguini, - in ‘Literature’ it is Perseguini, it should be corrected
Line 369 Xanthomonas citri pv. fuscans - pv. should not be in italic
Line 438 Dr.Brick – add a space
Line 537 Fusarium wilt race 1 and race 2 resistance – rather Fusarium wilt race 1 and race 4 resistance
Line 559 Phaseolus vulgaris - it should be in italic, similarly line 577, 581, 592, 604, 605, 610, and other places in the Literature for Latin names of plants or fungi
Line 574 verticillium and fusarium wilt – it should be Verticillium and Fusarium wilt
Line 617 fusarium oxysporum – it should be Fusarium oxysporum
Line 736 f. sp. Phaseoli – it should be f. sp. phaseoli
Author Response
Dear reviewer
I wanted to express my heartfelt thanks for your positive comments and careful review of our manuscript. Your kind words and comments regarding our work were truly motivating and heartening. Furthermore, we greatly appreciate the meticulousness and thoroughness with which you conducted your review. We have carefully revised and modified the manuscript in accordance with your recommendations.
Line 41 regions[3] – add a space. Such errors are numerous and should be corrected throughout the manuscript.
Thank you. We re-did all the references.
Figure 1 - It should be marked in Figure 1 where A is and where B is. Apart from that, what does FW (why not Fop) mean; race concerns the fungus Fusarium oxysporum ... and not the disease Fusarium wilt
Thank you. We re-did the Figure 1.
Line 153 Fusarium – it should be in italic, also in some other places in the manuscript where the pathogen is mentioned (of course, this does not apply to Fusarium wilt disease)
Thank you. We modified this.
Line 186 it is ‘SNP markers for race 4 resistance’ – it should be ‘SNP markers for race 1 resistance’
Thank you. We modified this.
Line 218 it is ‘with Fop race 2’ - it should be rather with Fop race 4
Thank you. We modified this.
Table 2, it is unclear which positions apply to race 1 and which to race 2 (see text in Line 175)
Thank you. We modified this.
Table 3 similarly as in Table 2, it is unclear which positions apply to race 1 and which to race 2 (see text in Lines 272-273)
Thank you. We modified this.
Line 290 table 3 – misconstructed table description – the verb is missing
Thank you. We modified this.
Line 357 Paulino et al.- in ‘Literature’ you write de Carvalho Paulino; this should be consistent
Thank you. We modified this.
Line 360 Leita˜o - in ‘Literature’ you write Leitao.
Thank you. We modified this.
Line 364 you write Persenguini, - in ‘Literature’ it is Perseguini, it should be corrected.
Thank you. We modified this.
Line 369 Xanthomonas citri pv. fuscans - pv. should not be in italic.
Thank you. We modified this.
Line 438 Dr.Brick – add a space
Thank you. We modified this.
Line 537 Fusarium wilt race 1 and race 2 resistance – rather Fusarium wilt race 1 and race 4 resistance
Thank you. We modified this.
Line 559 Phaseolus vulgaris - it should be in italic, similarly line 577, 581, 592, 604, 605, 610, and other places in the Literature for Latin names of plants or fungi
Thank you. We re-did all the references.
Line 574 verticillium and fusarium wilt – it should be Verticillium and Fusarium wilt
Thank you. We re-did all the references.
Line 617 fusarium oxysporum – it should be Fusarium oxysporum
Thank you. We re-did all the references.
Line 736 f. sp. Phaseoli – it should be f. sp. Phaseoli
Thank you. We re-did all the references.

Reviewer 2 Report
Summary
The manuscript explores Fusarium wilt resistance in common bean accessions using Genome-Wide Association Studies (GWAS) and Genomic Prediction (GP). It identifies highly resistant accessions, SNP markers, candidate genes, and demonstrates the potential of GP for predicting resistance.
This manuscript addresses an important agricultural issue, Fusarium wilt, which affects common bean crops. The study's use of both GWAS and GP is commendable, providing a comprehensive genetic analysis.
The study combines GWAS and GP, offering a holistic understanding of Fusarium wilt resistance in common beans. The identification of highly resistant accessions, SNP markers, and candidate genes provides valuable genetic resources for future breeding efforts. Demonstrating the potential for predictive accuracy in GP is crucial for practical applications in crop improvement. The manuscript is well-structured and clearly presented, making it accessible to a broad audience.
Major comments
- While the manuscript mentions data sources, providing direct links to datasets used in the analysis would enhance transparency and reproducibility.
- Some sections, especially those related to GWAS and GP, could benefit from additional statistical details to aid readers in understanding the methodologies.
- The manuscript could elaborate on the practical implications of the findings for farmers and crop improvement programs.
Overall, this manuscript contributes valuable insights into Fusarium wilt resistance in common beans, offering a strong foundation for future research and breeding efforts. Addressing the suggested comments will further enhance its quality and impact.
Minor comments
Abstract
- Consider defining abbreviations such as "SNP" (single nucleotide polymorphism), "GWAS" (genome-wide association study), "GP" (genomic prediction), "MAS" (marker-assisted selection), and "GS" (genomic selection) upon first use to enhance clarity for readers.
- In line 23, there's a missing space before "Chips."
- The sentences in lines 24 and 25 could be combined for smoother flow.
- Ensure consistent capitalization of chromosome names (e.g., "Pv03" should be consistent with "Pv04").
- Ensure consistency in verb tenses throughout the abstract. For example, you start with "This study aimed," which is in the past tense, but later in the abstract, you use "advances," which is in the present tense. Consistency in verb tenses is important.
- Consider rephrasing the sentence in line 27 for better clarity: "Moreover, 17 candidate resistance genes were found on chromosomes Pv03, Pv04, Pv05, Pv07, Pv08, Pv09, and Pv10." You could specify whether these genes are associated with Fop resistance or simply present on these chromosomes.
- Mention the total number of accessions (157) earlier in the abstract to provide context for the study.
Introduction
- Consider defining abbreviations such as "GWAS," "GP," "MAS," "SNP," and "PA" upon first use to make it easier for readers to understand.
Some sentences are long and complex, which may make them harder to follow. Consider breaking them into smaller sentences for improved readability.
- In line 39, you could specify what "tender green pods" are used for in those certain regions.
- In line 43, it would be clearer to mention the specific diseases common beans can mitigate the risk of (obesity, diabetes, certain cancers) earlier in the sentence.
- Consider rephrasing line 65 for better clarity. It's a bit wordy.
- In line 80, "in common bean" should be "in common beans" for subject-verb agreement.
- Consider adding a comma after "resistance against" in line 81 for better punctuation.
- In line 86, consider rephrasing the sentence for better flow.
- Make sure to include the specific references (e.g., [1, 2]) where appropriate to back up your statements.
Results
- "Phenotypic variation" and "Genomic prediction" should be consistently capitalized as section headings to maintain a consistent style.
- The subsection "2.3.2. SNP markers for race 4 resistance" appears to be missing a proper heading to introduce the content.
- In Table 2, there is an inconsistency in the use of "Chr" and "C hr" for chromosome abbreviations. Ensure uniformity (e.g., "Chr" is more common).
- Define abbreviations like "SNP," "LOD," and "QTL" when first used in the text to aid readers who may not be familiar with these terms.
- Table 3 formatting appears to be inconsistent. The columns are not well-aligned with the data they contain. Make sure that the table layout is clear and uniform.
- Ensure that measurements in tables (e.g., distances) are accompanied by units (e.g., bp for base pairs) where applicable.
- In Table 3, provide a brief explanation of what each gene's association with resistance means or its potential role in Fusarium wilt resistance.
- The figures should have clear and concise labels, including axis labels, to help readers understand the data presented.
- Check that figures are placed close to the relevant text or section in the manuscript for context.
- Use consistent terminology throughout the results section to maintain clarity.
Discussion
- The references in the discussion should be formatted consistently. Some references are presented in square brackets (e.g., [34]), while others are written in plain text (e.g., Paulino et al.). Use a consistent citation style, throughout the text.
- Some references in the text are incomplete and do not provide the necessary information to locate the source. Make sure to include the full citation information for each reference.
- In several places, there are missing periods at the end of sentences. Ensure that every sentence is properly punctuated.
- There are some typos and spelling errors, such as "Leita˜o" (should be "Leitão") and "phenotypic" (should be "phenotyping"). Carefully proofread the text to correct these errors.
- Some sentences are incomplete or lack clarity. For instance, in the sentence, "In this context, the realm of genomic prediction emerges as a viable panacea, capitalizing upon genomic," it is unclear what the sentence is trying to convey. Revise and complete such sentences for clarity.
- Some sentences are grammatically incorrect or awkwardly structured. For example, "These Fusarium resistance-associated regions are consistent with those reported by Paulino et al., who conducted a GWAS of Fusarium resistance in a core collection of 205 common bean genotypes..." The structure of this sentence can be improved for clarity.
- When introducing abbreviations like "SNP" (Single Nucleotide Polymorphism) and "GWAS" (Genome-Wide Association Study), it's good practice to provide their full forms the first time they are mentioned, followed by the abbreviation in parentheses.
- The use of the term "Fusarium wilt" is repeated multiple times in close proximity. Consider using synonyms or rephrasing to avoid redundancy.
- Some sentences, such as those discussing the genomic prediction models, are quite technical and may benefit from simplification or additional context to make them more accessible to a broader audience.
- Ensure consistent use of terminology throughout the discussion. For example, if you refer to "genomic prediction" in one place, continue using the same term instead of switching to "genomic information."
- Section headings should be properly formatted and numbered, like "3.1. Genetic diversity and population structure..."
- The numbering of sections, figures, and tables should be consistent. For example, if sections are numbered as "3.1, 3.2, 3.3," then figures and tables should follow the same numbering pattern.
- Ensure that all section headings, figure captions, and table titles are consistently capitalized according to a specific style guide.
- Some paragraphs may lack proper indentation, making the text harder to read. Ensure consistent indentation for each paragraph.
- When introducing abbreviations, consider adding a brief explanation or definition to make the text more understandable for readers who may not be familiar with the terms.
Materials and methods
- Ensure consistent formatting of references throughout the section. Some references are provided with links, while others are in standard citation format. Choose one format and stick to it consistently.
- Some references are not fully cited, and the publication details are missing. Make sure to include complete citation information for all sources.
- Section titles should be consistently capitalized. For example, "Principle component analysis (PCA)" should be "Principal Component Analysis (PCA)."
- Correct the spelling from "Principle component analysis" to "Principal Component Analysis."
- The paragraph starting with "Genome-wide association mapping" appears misaligned and should be adjusted for clarity.
- Incomplete Information on SNP Filtering**: When discussing SNP filtering, it would be helpful to briefly explain why SNPs with a missing rate >20%, heterogeneous >10%, and MAF <5% were excluded. This provides context for the readers.
- The abbreviation "ASI" is used without prior explanation. Provide a brief explanation of "ASI" when it is first introduced.
- Consistently use either "SNP" or "SNPs" throughout the section. For example, "SNPs" is used at the beginning, while "SNP markers" is used later. Choose one and stick with it.
- When providing web links to data sources or tools, ensure that they are properly formatted as hyperlinks.
- When "GP" is first mentioned in the context of Genomic Prediction (GP), provide a brief explanation of the abbreviation.
- When discussing different SNP sets (e.g., m32, r32, 4740), explain the significance or purpose of each set.
- Provide a brief explanation of the cross-validation process for Genomic Prediction to help readers understand how the data was partitioned and evaluated.
- Explain the abbreviation "GEBV" (Genomic Estimated Breeding Values) when it is first introduced.
- Ensure that references to supplementary figures and tables are properly formatted, and if possible, provide a clear reference to where readers can access this supplementary information.
- Double-check all references and ensure that they are complete, correctly formatted, and match the citation style you are using.
- Some sentences could be rephrased for clarity and flow, particularly in the sections about genetic diversity analysis and genomic prediction.
- Be consistent in how you mention the software or tools used for analysis. For example, use either "TASSEL 5" or "TASSEL5" consistently.
- When discussing LD calculations, briefly explain the importance of LD in genetic studies, if relevant.
- Carefully proofread the entire section for grammar, punctuation, and typos.
Conclusions
- In the conclusion, "GP" is used without prior explanation. Consider explaining the abbreviation "GP" (Genomic Prediction) when it is first introduced to ensure clarity for all readers.
- It would be helpful to briefly explain what a disease score of "3 or lower" signifies in terms of resistance. This clarification will make it easier for readers to interpret the results.
- When listing the five common bean accessions that were identified as resistant to both races, consider formatting them as a numbered or bulleted list for clarity.
- Ensure consistent use of chromosome abbreviations. For example, you've used both "Pv9" and "Pv9." Choose one format and stick with it throughout the conclusion.
- It's unclear what "PA (r100_value)" represents without prior explanation. Consider providing a brief explanation or reference to where this metric is defined in the text.
- The conclusion mentions the identified SNP markers and associated genes as valuable genetic resources, but it doesn't explicitly state how they can be utilized. Consider adding a sentence or two about their potential applications in breeding programs or research.
- Conclusions should summarize the key findings of the study. Consider starting the conclusion with a clear statement of the main findings before delving into details.
- Ensure that each sentence is clear and concise, avoiding overly complex language or terminology that may confuse readers.
- Carefully proofread the manuscript for grammar, punctuation, and typographical errors to enhance its overall readability.
Moderate editing of English language required.
Author Response
Dear reviewer
I wanted to express my heartfelt thanks for your positive comments and careful review of our manuscript. Your meticulous review and constructive feedback were invaluable. We have diligently addressed each of your suggestions and believe that the manuscript has greatly improved as a result. Your expertise and guidance have been instrumental in enhancing the quality of our research.
We are grateful for your time and dedication to maintaining scholarly excellence. If you ever wish to collaborate or review our future work, please do not hesitate to reach out. Your insights are highly valued.
Major comments
- While the manuscript mentions data sources, providing direct links to datasets used in the analysis would enhance transparency and reproducibility.
Reply: Thank you for the kindness reminding! The phenotype was fully offered in the Supplementary Materials. The genotype was offered by the link of :https://datadryad.org/stash/dataset/doi:10.25338/B8KP45, which was also shown at the part of Materials and Method : 4.2. Genotyping and SNP Selection.
- Some sections, especially those related to GWAS and GP, could benefit from additional statistical details to aid readers in understanding the methodologies.
Reply: Thank you for the valuable suggestion. We added two paragraphs at the end of the each discussion part of GWAS and GP.
- The manuscript could elaborate on the practical implications of the findings for farmers and crop improvement programs.
Reply: Thank you for the valuable advice. We added an elaboration to the Conclusions part.
Minor comments
Abstract
- Consider defining abbreviations such as "SNP" (single nucleotide polymorphism), "GWAS" (genome-wide association study), "GP" (genomic prediction), "MAS" (marker-assisted selection), and "GS" (genomic selection) upon first use to enhance clarity for readers.
Reply: Thank you. We added the full name of those abbreviations upon first use.
- In line 23, there's a missing space before "Chips."
Reply: We modified this.
- The sentences in lines 24 and 25 could be combined for smoother flow.
Reply: Thank you. We modified this.
- Ensure consistent capitalization of chromosome names (e.g., "Pv03" should be consistent with "Pv04").
Reply: Thank you. We modified this.
- Ensure consistency in verb tenses throughout the abstract. For example, you start with "This study aimed," which is in the past tense, but later in the abstract, you use "advances," which is in the present tense. Consistency in verb tenses is important.
Reply: Thank you. We modified this.
- Consider rephrasing the sentence in line 27 for better clarity: "Moreover, 17 candidate resistance genes were found on chromosomes Pv03, Pv04, Pv05, Pv07, Pv08, Pv09, and Pv10." You could specify whether these genes are associated with Fop resistance or simply present on these chromosomes.
Reply: Thank you. We re-wrote this part.
- Mention the total number of accessions (157) earlier in the abstract to provide context for the study.
Reply: Thank you. We re-wrote this part.
Introduction
- Consider defining abbreviations such as "GWAS," "GP," "MAS," "SNP," and "PA" upon first use to make it easier for readers to understand.
Reply: Thank you. We added the full name and some explanations of those abbreviations upon first use.
Some sentences are long and complex, which may make them harder to follow. Consider breaking them into smaller sentences for improved readability.
Reply: Thank you. We modified some sentences.
- In line 39, you could specify what "tender green pods" are used for in those certain regions.
Reply: Thank you. We re-wrote this part.
- In line 43, it would be clearer to mention the specific diseases common beans can mitigate the risk of (obesity, diabetes, certain cancers) earlier in the sentence.
Reply: Thank you. We re-wrote this part.
- Consider rephrasing line 65 for better clarity. It's a bit wordy.
Reply: Thank you. We re-wrote this part.
- In line 80, "in common bean" should be "in common beans" for subject-verb agreement.
Reply: Thank you. We modified this.
- Consider adding a comma after "resistance against" in line 81 for better punctuation.
Reply: Thank you. We re-wrote this part.
- In line 86, consider rephrasing the sentence for better flow.
Reply: Thank you. We re-wrote this part.
- Make sure to include the specific references (e.g., [1, 2]) where appropriate to back up your statements.
Reply: Thank you, we added one reference.
Results
- "Phenotypic variation" and "Genomic prediction" should be consistently capitalized as section headings to maintain a consistent style.
Reply: Thank you. We modified these issues.
- The subsection "2.3.2. SNP markers for race 4 resistance" appears to be missing a proper heading to introduce the content.
Reply: Thank you. We modified this.
- In Table 2, there is an inconsistency in the use of "Chr" and "C hr" for chromosome abbreviations. Ensure uniformity (e.g., "Chr" is more common).
Reply: Thank you. We modified this.
- Define abbreviations like "SNP," "LOD," and "QTL" when first used in the text to aid readers who may not be familiar with these terms.
Reply: Thank you. We modified this.
- Table 3 formatting appears to be inconsistent. The columns are not well-aligned with the data they contain. Make sure that the table layout is clear and uniform.
Reply: Thank you. We modified this.
- Ensure that measurements in tables (e.g., distances) are accompanied by units (e.g., bp for base pairs) where applicable.
Reply: Thank you. We modified this.
- In Table 3, provide a brief explanation of what each gene's association with resistance means or its potential role in Fusarium wilt resistance.
Reply: Thank you. We modified this.
- The figures should have clear and concise labels, including axis labels, to help readers understand the data presented.
Reply: Thank you. We modified this.
- Check that figures are placed close to the relevant text or section in the manuscript for context.
Reply: Yes, they are.
- Use consistent terminology throughout the results section to maintain clarity.
Reply: Yes, we checked them.
Discussion
- The references in the discussion should be formatted consistently. Some references are presented in square brackets (e.g., [34]), while others are written in plain text (e.g., Paulino et al.). Use a consistent citation style, throughout the text.
Reply: We followed the guideline of the Journal requires.
- Some references in the text are incomplete and do not provide the necessary information to locate the source. Make sure to include the full citation information for each reference.
Reply: Thank you. We re-did the references.
- In several places, there are missing periods at the end of sentences. Ensure that every sentence is properly punctuated.
Reply: Thank you. We modified this.
- There are some typos and spelling errors, such as "Leita˜o" (should be "Leitão") and "phenotypic" (should be "phenotyping"). Carefully proofread the text to correct these errors.
Reply: Thank you. We modified this.
- Some sentences are incomplete or lack clarity. For instance, in the sentence, "In this context, the realm of genomic prediction emerges as a viable panacea, capitalizing upon genomic," it is unclear what the sentence is trying to convey. Revise and complete such sentences for clarity.
Reply: Thank you. We re-wrote this part.
- Some sentences are grammatically incorrect or awkwardly structured. For example, "These Fusarium resistance-associated regions are consistent with those reported by Paulino et al., who conducted a GWAS of Fusarium resistance in a core collection of 205 common bean genotypes..." The structure of this sentence can be improved for clarity.
Reply: Thank you. We re-wrote this part.
- When introducing abbreviations like "SNP" (Single Nucleotide Polymorphism) and "GWAS" (Genome-Wide Association Study), it's good practice to provide their full forms the first time they are mentioned, followed by the abbreviation in parentheses.
Reply: Thank you. We added the full name of those abbreviations upon first use in the manuscript.
- The use of the term "Fusarium wilt" is repeated multiple times in close proximity. Consider using synonyms or rephrasing to avoid redundancy.
Reply: Thank you. We modified this.
- Some sentences, such as those discussing the genomic prediction models, are quite technical and may benefit from simplification or additional context to make them more accessible to a broader audience.
Reply: Thank you. We re-wrote this part.
- Ensure consistent use of terminology throughout the discussion. For example, if you refer to "genomic prediction" in one place, continue using the same term instead of switching to "genomic information."
Reply: Thank you. We re-wrote this part.
- Section headings should be properly formatted and numbered, like "3.1. Genetic diversity and population structure..."
Reply: Thank you. We modified this.
- The numbering of sections, figures, and tables should be consistent. For example, if sections are numbered as "3.1, 3.2, 3.3," then figures and tables should follow the same numbering pattern.
Reply: Thank you. We have checked this.
- Ensure that all section headings, figure captions, and table titles are consistently capitalized according to a specific style guide.
Reply: Thank you. We have checked this.
- Some paragraphs may lack proper indentation, making the text harder to read. Ensure consistent indentation for each paragraph.
Reply: Thank you. We have checked this.
- When introducing abbreviations, consider adding a brief explanation or definition to make the text more understandable for readers who may not be familiar with the terms.
Reply: Thank you. We re-wrote some of these.
Materials and methods
- Ensure consistent formatting of references throughout the section. Some references are provided with links, while others are in standard citation format. Choose one format and stick to it consistently.
Reply: Thank you. We re-did the references.
- Some references are not fully cited, and the publication details are missing. Make sure to include complete citation information for all sources.
Reply: Thank you. We re-did the references.
- Section titles should be consistently capitalized. For example, "Principle component analysis (PCA)" should be "Principal Component Analysis (PCA)."
Reply: Thank you. We modified this.
- Correct the spelling from "Principle component analysis" to "Principal Component Analysis."
Reply: Thank you. We modified this.
- The paragraph starting with "Genome-wide association mapping" appears misaligned and should be adjusted for clarity.
Reply: Thank you. We modified this.
- Incomplete Information on SNP Filtering**: When discussing SNP filtering, it would be helpful to briefly explain why SNPs with a missing rate >20%, heterogeneous >10%, and MAF <5% were excluded. This provides context for the readers.
Reply: Thank you. We re-wrote this part.
- The abbreviation "ASI" is used without prior explanation. Provide a brief explanation of "ASI" when it is first introduced.
Reply: Thank you. We re-wrote this part.
- Consistently use either "SNP" or "SNPs" throughout the section. For example, "SNPs" is used at the beginning, while "SNP markers" is used later. Choose one and stick with it.
Reply: Thank you. We modified this.
- When providing web links to data sources or tools, ensure that they are properly formatted as hyperlinks.
Reply: Thank you. We have checked it.
- When "GP" is first mentioned in the context of Genomic Prediction (GP), provide a brief explanation of the abbreviation.
Reply: Thank you. There is a brief explanation in the introduction part.
- When discussing different SNP sets (e.g., m32, r32, 4740), explain the significance or purpose of each set.
Reply: Thank you. We re-wrote this part.
- Provide a brief explanation of the cross-validation process for Genomic Prediction to help readers understand how the data was partitioned and evaluated.
Reply: Thank you. We re-wrote this part.
- Explain the abbreviation "GEBV" (Genomic Estimated Breeding Values) when it is first introduced.
Reply: Thank you. We re-wrote this part.
- Ensure that references to supplementary figures and tables are properly formatted, and if possible, provide a clear reference to where readers can access this supplementary information.
Reply: Thank you. The supplementary figures and tables applied in this study are all available
- Double-check all references and ensure that they are complete, correctly formatted, and match the citation style you are using.
Reply: Thank you. We re-did the references.
- Some sentences could be rephrased for clarity and flow, particularly in the sections about genetic diversity analysis and genomic prediction.
Reply: Thank you. We re-wrote couple of them.
- Be consistent in how you mention the software or tools used for analysis. For example, use either "TASSEL 5" or "TASSEL5" consistently.
Reply: Thank you. We modified this.
- When discussing LD calculations, briefly explain the importance of LD in genetic studies, if relevant.
Reply: Thank you. We re-wrote this part.
- Carefully proofread the entire section for grammar, punctuation, and typos.
Reply: Thank you. We have checked it.
Conclusions
- In the conclusion, "GP" is used without prior explanation. Consider explaining the abbreviation "GP" (Genomic Prediction) when it is first introduced to ensure clarity for all readers.
Reply: Thank you. There is a brief explanation in the introduction part.
- It would be helpful to briefly explain what a disease score of "3 or lower" signifies in terms of resistance. This clarification will make it easier for readers to interpret the results.
Reply: Thank you. We re-wrote this part.
- When listing the five common bean accessions that were identified as resistant to both races, consider formatting them as a numbered or bulleted list for clarity.
Reply: Thank you. We re-wrote this part.
- Ensure consistent use of chromosome abbreviations. For example, you've used both "Pv9" and "Pv9." Choose one format and stick with it throughout the conclusion.
Reply: Thank you. We modified this.
- It's unclear what "PA (r100_value)" represents without prior explanation. Consider providing a brief explanation or reference to where this metric is defined in the text.
Reply: Thank you. The r100 means 100-time repeats of the calculations. Commonly, we use this to label the PA from different repeat time. Considering there were no difference repeat applied in this study, we will not use this label in this manuscript to reduce the confusion.
- The conclusion mentions the identified SNP markers and associated genes as valuable genetic resources, but it doesn't explicitly state how they can be utilized. Consider adding a sentence or two about their potential applications in breeding programs or research.
Reply: Thank you. We added a brief summary at the end.
- Conclusions should summarize the key findings of the study. Consider starting the conclusion with a clear statement of the main findings before delving into details.
Reply: Thank you. We re-wrote this part.
- Ensure that each sentence is clear and concise, avoiding overly complex language or terminology that may confuse readers.
Reply: Thank you. We re-wrote this part.
- Carefully proofread the manuscript for grammar, punctuation, and typographical errors to enhance its overall readability.
Reply: Thank you. We re-wrote some sentences.

Round 2
Reviewer 2 Report
The authors have effectively addressed the reviewer’s feedback and made necessary revisions to their manuscript. I am highly satisfied with the revised version. I wholeheartedly recommend this manuscript for publication in its current form. Congratulations on this excellent publication!